# The Langmuir Monolayer as a Model Membrane System for Studying the Interactions of Poly(Butyl Cyanoacrylate) Nanoparticles with Phospholipids at the Air/Water Interface

**DOI:** 10.3390/membranes14120254

**Published:** 2024-12-02

**Authors:** Georgi Yordanov, Ivan Minkov, Konstantin Balashev

**Affiliations:** 1Department of Inorganic Chemistry, Faculty of Chemistry and Pharmacy, Sofia University St. Kliment Ohridski, 1 “James Bourchier” Blvd., 1164 Sofia, Bulgaria; nhgy@chem.uni-sofia.bg; 2Department of Chemistry, Biochemistry, Physiology, and Pathophysiology, Faculty of Medicine, Sofia University, 1407 Sofia, Bulgaria; ilminkov@med.uni-sofia.bg; 3Institute of Physical Chemistry, Bulgarian Academy of Sciences, Acad. G. Bonchev Str., bl. 11, 1113 Sofia, Bulgaria; 4Department of Physical Chemistry, Faculty of Chemistry and Pharmacy, Sofia University St. Kliment Ohridski, 1 “James Bourchier” Blvd., 1164 Sofia, Bulgaria

**Keywords:** poly(butyl cyanoacrylate) nanoparticles, Pluronic 68, Langmuir monolayers, atomic force microscopy (AFM)

## Abstract

Poly(butyl cyanoacrylate) (PBCA) nanoparticles have numerous applications, including drug and gene delivery, molecular imaging, and cancer therapy. To uncover the molecular mechanisms underlying their interactions with cell membranes, we utilized a Langmuir monolayer as a model membrane system. This approach enabled us to investigate the processes of penetration and reorganization of PBCA nanoparticles when deposited in a phospholipid monolayer subphase. Atomic force microscopy (AFM) was employed to visualize Langmuir–Blodgett (LB) films of these nanoparticles. Additionally, we examined the state of a monolayer of Pluronic F68, a stabilizer of PBCA nanoparticles in suspension, by measuring the changes in relative surface area and surface potential over time in the barostatic regime following PBCA suspension spreading. Based on these findings, we propose a molecular mechanism for nanoparticle reorganization at the air–water interface.

## 1. Introduction

In recent years, submicron and nano-sized particles with diverse compositions and structures have opened a broad scientific horizon in modern technologies and the pharmaceutical industry for the development of new materials, biodegradable drug carriers, components for microelectronics, catalysts in fine organic synthesis, various nanoemulsions, and more [1,2,3,4,5]. Due to their small size—10 to 100 times smaller than a typical human body cell (5–10 μm)—these particles are well-suited as carriers for various drugs and biologically active molecules, including proteins, hormones, DNA, and RNA [6,7,8].

Poly(n-butyl cyanoacrylate) (PBCA) nanoparticles, composed of a polymer with a medium-length alkyl side chain, offer numerous advantages, including low toxicity and biodegradability. These properties have made PBCA nanoparticles one of the most widely used biocompatible drug carriers in the experimental development of polymer-based nanocarriers for over four decades since their discovery in the late 1970s [9,10,11,12,13]. Recent studies have shown that surfactant-coated PBCA nanoparticles can interact with the vascular endothelium, enhancing transcytosis and facilitating drug delivery to the brain [10]. Moreover, cyclic peptide- and Pluronic-modified PBCA nanoparticles have been shown to increase drug penetration efficacy through blood–brain barrier models [11,14].

Drug-loaded PBCA nanoparticles are typically prepared by dispersing the drug during the polymerization process; however, a significant challenge of this method lies in achieving highly stable nanoparticle dispersions using biocompatible and non-toxic stabilizers. A promising solution to enhance the colloidal stability of PBCA nanoparticles is the use of the triblock copolymer poly(ethylene oxide)-poly(propylene oxide)-poly(ethylene oxide), commonly known as Pluronic F68 (poloxamer 188) [15,16].

Poly(n-butyl cyanoacrylate) (PBCA) nanoparticles, synthesized from a polymer with a medium-length alkyl side chain, offer several advantages, including low toxicity and biodegradability, making them one of the most widely used biocompatible drug carriers [13]. The drug-carrying PBCA nanoparticles are conventionally prepared by dispersing the drug during the polymerization of the nanoparticles; however, there are drawbacks to their use, mainly because some substances are not stable under the polymerization conditions, and the highly reactive alkyl cyanoacrylate monomer can irreversibly deactivate some drugs. Additionally, polymerization conditions significantly influence the size distribution of the nanoparticles and the molecular weight of the resulting polymer.

To overcome these drawbacks, nanoprecipitation is commonly used to obtain pure drug carriers. This method enables the incorporation of hydrophobic and chemically unstable drug substances by avoiding contact with the highly reactive monomer, thus preventing drug deactivation [17].

A notable model for the incorporation and release of drug substances from such carriers is the alkylating anticancer agent chlorambucil, a lipophilic drug used to treat chronic lymphocytic leukemia, lymphomas, and other malignancies. Chlorambucil contains electrophilic reactive groups that can alkylate nucleic acids and proteins at their nucleophilic centers, forming intrachain and interchain bonds. These changes disrupt replication, transcription, and translation processes within the cell, ultimately leading to the damage or death of neoplastic cells; however, chlorambucil’s use is limited because of its chemical instability and toxic effects on the human body. A promising approach to address these limitations is incorporating the drug into PBCA nanoparticles and stabilizing the drug-carrier complex with the tri-block copolymer poly(ethylene oxide)-poly(propylene oxide)-poly(ethylene oxide), commonly known as Pluronic F68 [16].

While the production of nontoxic, biocompatible PBCA drug carriers is essential, a more intriguing aspect is understanding the mechanisms of interaction between these nanoparticles and cell membranes, as well as how they cross the cell barrier to enter the intracellular matrix. A reliable approach to studying the behavior and reorganization of PBCA nanoparticles containing encapsulated drugs at the cell membrane interface is to use a Langmuir monolayer of phospholipids spread at the air/water interface as a simplified and convenient cell membrane model. This approach has already been successfully applied to study the interfacial interaction of lipid nanocapsules (LNCs) with a Dipalmitoylphosphatidylcholine (DPPC) monolayer at the air/water interface [18,19,20,21,22]. A monolayer of the saturated phospholipid DPPC is often preferred as a simplified model of the cell membrane, primarily because DPPC, a phosphatidylcholine, is one of the most common components of pulmonary surfactants in mammalian cell membranes [23]. It forms insoluble monolayers at the air/water interface with well-defined phase transitions, closely resembling those of biological membranes and mimicking the lateral interactions within them [24,25,26]. Moreover, drug insertion into preformed DPPC monolayers at the air/water interface provides an accurate model for drug penetration through biological membranes [27,28,29,30,31].

This paper aims to investigate the mechanisms of destabilization and reorganization of polymer PBCA nanoparticles stabilized with Pluronic F68 in a colloidal suspension at the air/water interface, as well as their interactions with preformed DPPC monolayers. To achieve this, we employed a classical approach by measuring surface pressure and surface potential isotherms and monitoring changes in these interfacial thermodynamic properties of the Langmuir monolayer over time. Additionally, the monolayer studies at the air/water interface were complemented with Atomic Force Microscopy (AFM) imaging of Langmuir–Blodgett (LB) films, transferred from the air/water interface onto mica solid support.

## 2. Materials and Methods

### 2.1. Materials

Butyl cyanoacrylate (BCA), used as a monomer for nanoparticle synthesis, was purchased from Special Polymers Ltd. (Surrey, UK). The colloidal stabilizer, poloxamer 188 (Pluronic F-68; a triblock copolymer of poly(ethylene oxide)-poly(propylene oxide)-poly(ethylene oxide) with a molecular weight of 8400), and polysorbate 80 (Tween 80; polyoxymethylene (20) sorbitan monooleate) were purchased from Sigma-Aldrich (Merck Bulgaria EAD, Sofia, Bulgaria). The phospholipid 1,2-dipalmitoyl-sn-glycero-3-phosphocholine monohydrate (DPPC), used for monolayer studies, was obtained from Avanti Polar Lipids (Merck Bulgaria EAD, Sofia, Bulgaria). All reagents were of analytical grade, and double-distilled water was used in all experiments.

### 2.2. Methods

Colloidal PBCA nanoparticles were prepared by controlled emulsion polymerization of butyl cyanoacrylate, as described elsewhere [16]. Briefly, the polymerization medium was prepared by dissolving 500 mg of Pluronic F68 and 400 mg of citric acid in 200 mL of distilled water. Then, 2 mL of butyl cyanoacrylate monomer was added dropwise to the medium under vigorous stirring at 600 rpm. Within the first 10 min, the emulsion turned milky white, and polymerization proceeded for six hours. The pH of the resulting dispersion was adjusted to 5.6 by adding 4 mL of 1M NaOH. Next, the polymerized dispersion was centrifuged at 14,500 rpm for 15 min and washed twice with double-distilled water. The resulting precipitate was dried under vacuum, yielding a fine white powder used for nanoparticle preparation.

For PBCA nanoparticle preparation, the precipitation medium was prepared with 20 mg of Pluronic F68 as a colloidal stabilizer and 20 mg of citric acid dissolved in 10 mL of 5% glucose solution. Then, 50–100 mg of pre-synthesized PBCA was dissolved in 5 mL of acetone and added dropwise to the nanoprecipitation medium under vigorous stirring at 600 rpm, allowing acetone to evaporate as the suspension was left in a fume hood for 5 h. Residual acetone was removed by vacuum evaporation. Nanoparticle concentrations were determined gravimetrically by repeated centrifugation and washing of particles from an aliquot of the dispersion, followed by drying of the sediment to a constant mass in pre-weighed Eppendorf tubes in a vacuum desiccator. Water dispersions of PBCA nanoparticles with concentrations of 6–8 mg/mL were prepared, as well as PBCA nanoparticle solutions in chloroform with concentrations of 1.5 ± 0.2 mg/mL.

It should be noted that the prepared PBCA is soluble in acetone but not in water; therefore, Pluronic F68, a triblock copolymer of poly(ethylene oxide)-poly(propylene oxide)-poly(ethylene oxide) or shortly (EO₇₆-PO₂₉-EO₇₆) [32], was used as a colloidal stabilizer to stabilize the nanoparticles in the water phase. This polymer has a molecular weight of 8400 and consists of three parts: two hydrophilic EO blocks and a central hydrophobic PO block (Figure 1a). This structure enables Pluronic molecules to adsorb on the surface of PBCA nanoparticles, creating a hydrophilic shell around them that prevents aggregation. A possible molecular organization of Pluronic F68 molecules surrounding the PBCA nanoparticle is illustrated in Figure 1b. Since PBCA nanoparticles are inherently hydrophobic, Pluronic F68, acting as a surfactant, serves as a stabilizing agent and is expected to form a shell around individual nanoparticles. In this configuration, the hydrophobic regions of Pluronic F68 interact with the nanoparticle surface, while the hydrophilic regions face the aqueous phase. Assuming that Pluronic F68 adsorption on the nanoparticle surface is similar to its behavior at the air/water interface (as discussed further), we can estimate the number of molecules involved. Given a mean nanoparticle radius of approximately 100 nm (1000 Å), the surface area of a single PBCA nanoparticle can be calculated as $S_{NP}=4\pi r^{2}=1.25*10^{7}{Å}^{2}$. For a closely packed monolayer of Pluronic F68 molecules at the air/water interface, each molecule occupies an area of: $S_{Pluronic\;F68}=500\;{Å}^{2}\;\mathrm{per}\;\mathrm{molecule}$, Thus, the number of Pluronic F68 molecules that a single PBCA nanoparticle can accommodate is estimated as: $n=\frac{S_{NP}}{S_{Pluronic\;F68}}=25000\;Pluronic\;F68\;molecules$.

The PBCA nanoparticles were characterized by atomic force microscopy (AFM). Sample preparation for AFM imaging involved depositing a freshly synthesized PBCA nanoparticle solution onto a freshly cleaved mica surface before the addition of Pluronic F68. Freshly cleaved Grade V-4 Muscovite mica sheets (10 × 10 mm) from SPI Supplies (Structure Probe, Inc.,West Chester, PA, USA) were used for the deposition of approximately 100 μL of nanoparticle solution. These mica sheets were also used in LB experiments. After a 10-min incubation, samples were gently dried with nitrogen gas for 5 min.

AFM imaging was performed on a NanoScope MultiMode V system (Bruker Nano GmbH, Berlin, Germany) in tapping mode in air at room temperature. The system uses a stationary probe oscillated vertically by a piezoelectric stack, while the sample, mounted on a metal puck magnetically attached to the scanner tube, is translated horizontally. The probe detects sample surface information through the deflection of the cantilever as it encounters the surface, revealing vertical height and other features of the deposited material. Silicon cantilevers (Tap 300 Al-G, Budget Sensors, Innovative Solutions Ltd., Sofia, Bulgaria) with a 30 nm-thick aluminum reflex coating were used, with a spring constant range of 1.5–15 N/m and a resonance frequency of 150 ± 75 kHz. The tip radius was under 10 nm.

Before imaging, samples were thoroughly dried with N₂ gas. The scans were conducted at a scanning rate of 1 Hz, capturing images in both height and phase modes at 512 × 512-pixel resolution and saved in JPEG format. Multiple areas along the mica sheets were scanned, and all images were flattened using Nanoscope software (v.7.30). These imaging settings were consistently applied across all AFM experiments.

Figure 2 shows typical AFM images of PBCA nanoparticles deposited on the mica surface in 2D topography format, with a scanned area of 5 × 5 μm^2^ and a z-scale of 300 nm (Figure 2a), where the arrangement of the polymer nanoparticles in the X-Y plane is clearly distinguishable.

In the other images, the scanned area along the XY-plane was reduced to 2 × 2 μm^2^, and the images were presented in both 2D (Figure 2b) and 3D (Figure 2c) formats. The analysis of nanoparticle size was performed through section analysis across individual nanoparticles, measuring their diameters while considering a well-known artifact in AFM metrology known as the convolution (rounding) effect. This effect arises from the interaction between the AFM tip and the surface, as well as the size of the tip’s radius of curvature [33]. The histogram for the size distribution and the average size of the nanoparticles derived from the section analysis is presented in Figure 2d. The estimated diameter of the PBCA nanoparticles was d = 200 nm, consistent with reported results from scanning electron microscopy (SEM) and dynamic light scattering (DLS) measurements [17].

Phospholipid monolayers at the air/water interface are convenient and well-defined biological membrane models [24,34,35,36,37]. They are formed after the spreading of phospholipids from their volatile organic solutions on a liquid surface in a Langmuir trough (Figure 3a). Due to the thermal motion of phospholipid molecules that is constrained within the plane of the phospholipid monolayer at the air/water interface and the difference in the surface tensions of the liquid phase and that of the phospholipid monolayer, a surface pressure (π) arises. This is an experimentally measurable quantity and the main characteristic of the thermodynamic state of the monolayer. Hence, using the Langmuir trough, equipped with movable barriers, by controlling the area of the monolayer, one can obtain the dependence of the surface pressure (π) versus the area per one phospholipid molecule (${Å}^{2}/\mathrm{molecule}$), i.e., surface pressure/area isotherm.

Due to the thermal motion of phospholipid molecules that is constrained within the plane of the phospholipid monolayer at the air/water interface and the difference in the surface tensions of the liquid phase and that of the phospholipid monolayer, a surface pressure (π) arises. This is an experimentally measurable quantity and the main characteristic of the thermodynamic state of the monolayer. Hence, using the Langmuir trough, equipped with movable barriers, by controlling the area of the monolayer, one can obtain the dependence of the surface pressure (π) versus the area per one phospholipid molecule (${Å}^{2}/\mathrm{molecule}$), i.e., surface pressure/area isotherm.

In Figure 3b (blue curve), a typical DPPC isotherm is presented. At the largest areas per molecule, the monolayer exists in a molecular state where individual DPPC molecules are separated by distances too great for any force interactions among them to occur. This thermodynamic state is considered a 2D gas. Further compression of the monolayer leads to the appearance of different phases or states, which depend on surface concentration (or surface pressure), temperature, and molecular structure. It is now generally accepted that a DPPC monolayer can exist in four distinct pure phases below the critical temperature (~40 °C): gas (G), liquid expanded (LE), liquid condensed (LC), and solid or crystalline (S) phases (monolayer states) [34]. From the π-A curve (Figure 3b), under these conditions, the area per phospholipid molecule can be determined; for a DPPC monolayer, this area is 49 Å^2^.

On the other hand, at the air/water interface between the two phases, the dipole moments of the phospholipid molecules remain partially uncompensated, resulting in a potential difference known as the Volta potential. This potential is defined as the work required to transfer a unit charge from infinity to the interface and equals the difference between the potentials of the water substrate and the monomolecular layer spread at the air/water interface. The Volta potential is proportional to the effective dipole density (μ) for an uncharged monolayer and represents the main electrical characteristic of the monolayer. In a continuum approximation, the Helmholtz equation gives:(1)$$∆V=\frac{4\pi \;(\bar{\mu }\;n)}{\varepsilon },$$
where ε is the dielectric constant (assuming it is unity), n corresponds to the number of molecules at the surface (or the surface concentration Γ), and $\bar{\mu }$ is the vertical component of the dipole moment of the molecules. Measuring ΔV when the area of the monolayer changes provides information about the change in the perpendicular component of the dipole moments of the phospholipid molecules, reflecting structural changes in the monolayer [38]. In Figure 3b (red curve), a typical surface potential versus area isotherm of the DPPC monolayer is also presented, where all states of the monolayer during compression are distinguishable, particularly the phase transition from G to L, which is undetectable on the surface pressure/area isotherm.

In this study, the isotherms of Pluronic F68 monolayers were measured. The monolayers were spread on a water subphase in a Langmuir film balance KSV 2200 (KSV Instruments Ltd., Helsinki, Finland) with a maximum available trough area of 476 cm^2^. A Hamilton microsyringe was used to spread 10 μL of Pluronic F68 from a chloroform solution at the water interface. Both surface potential (ΔV) and surface pressure (Δπ) were simultaneously measured, as shown in Figure 3a. For surface potential measurements, the ionization method was employed, which required a gold probe coated with an emitter of ^241^Am (approximately 750 μCi) positioned above the surface and a reference calomel electrode, both connected to the KP511 electrometer (Kryona Ltd., Sofia, Bulgaria). The surface potential of the pure aqueous surface was approximately −150 mV, as reported elsewhere [39], and fluctuated over a 30-min period. The reproducibility of the initial surface potential value was within ±15 mV. Once the air–water surface potential stabilized, the monolayer spreading was carried out. Real-time data acquisition was managed by a personal computer equipped with user-specific hardware and software.

After the monolayer spread, the surface pressure initially fluctuated around 0.1 mNm^−1^ and stabilized after approximately 10–15 min. The monolayer was then compressed at a constant velocity of U_b_ = 50 cm^2^min^−1^. Under these conditions, the isotherms were highly reproducible. At least three measurements were performed for this and all other experiments in the study to confirm reproducibility.

During the reorganization and possible penetration of PBCA nanoparticles into the DPPC monolayer, both the increase in surface area (ΔA) and the evolution of surface potential (ΔV) over time (t) were simultaneously measured at a constant surface pressure (π). A “zero-order” Langmuir trough containing a reaction compartment with an area of 50 cm^2^ connected to a reservoir compartment (area 260 cm^2^) by a 0.5 cm wide channel was utilized (Figure 3a).

After spreading over the maximum available area (A = 310 cm^2^), the DPPC monolayer was compressed at a constant velocity of U_b_ = 100 cm^2^min^−1^ to a set endpoint surface pressure, after which the barostatic mode was activated. Ten minutes later, 50 μL of PBCA nanoparticle dispersion was injected into the reaction compartment containing a 0.15 M NaCl solution. Due to the penetration of PBCA nanoparticles into the DPPC monolayer and their reorganization at the air/water interface, the surface pressure increased, with the surface pressure barostat adjusting by barrier displacement, thereby altering the surface area (ΔA). The final concentration of PBCA in the reaction compartment was $C_{PBCA}=0.01\;\pm \;0.002\;\mathrm{mg}/\mathrm{mL}$. The bulk in the reaction compartment was continuously stirred at 250 rpm using a magnetic rod, and the reservoir compartment’s aqueous subphase was also 0.15 M NaCl solution.

The monolayers were transferred from the air/water interface onto mica supports via Langmuir–Blodgett (LB) deposition. For this, the solid support was dipped vertically into the subphase and then moved upward through the monolayer. The mica plates were removed from the subphase approximately 50 min after the injection of the PBCA nanoparticles at a transfer rate of 5 mm/min. Before AFM imaging, the LB film was thoroughly dried with a gentle nitrogen gas flow for about 10–15 min and then imaged as described above.

## 3. Results and Discussion

To investigate the states and properties of Pluronic F68 monolayers, a preliminary and essential experimental step involved performing isotherm measurements at the water/air interface at 25 °C, using a 0.15 M NaCl solution as a substrate. The results obtained provide valuable information on the interfacial thermodynamic state of the monolayers under investigation and can also be confirmed and compared with previously published data [40].

Figure 4 presents the surface pressure/area isotherm (blue curve) and the surface potential/area isotherm (red curve) on the same graph.

Our experimental data start from areas of 6500 Å^2^/molecule, tracking the progression of the surface pressure/area isotherm curve during the compression of the Pluronic 68 monolayer to areas as small as 500 Å^2^/molecule. Technically, we were able to monitor the surface potential only down to areas of about 1800 Å^2^/molecule; however, using previously published data [40], it can be inferred that for areas between 8000 Å^2^/molecule and 6500 Å^2^/molecule, the monolayer is most likely in a gaseous-liquid expanded (G-LE) state, as schematically illustrated in Figure 4. Within this range, the surface pressure gradually and slowly increases while the surface potential also rises because of the freedom of the polymer’s monomer units at the air/water interface, which contributes to changes in the vertical component of the total dipole moment.

Further compression of the monolayer from 6500 Å^2^/molecule to approximately 3000 Å^2^/molecule results in the appearance of a liquid-expanded (LE) phase, followed by a sloping transition plateau that corresponds to a liquid-condensed one (LC1) phase. Strong evidence for this monolayer transition is provided by the surface potential isotherm, which exhibits a plateau in this region, as shown in Figure 4. In this state, the polymer molecules of Pluronic F68 are closely packed at the air/water interface.

At a surface pressure of approximately 10.5 mNm^−1^ and a corresponding area of 2700 Å^2^/molecule, assuming that the surface areas occupied by the ethylene oxide (EO) and propylene oxide (PO) monomer units of Pluronic F68 are approximately equal, the average area per monomer unit can be estimated. Dividing the area of the closely packed monolayer (2700 Å^2^/molecule) by the total number of monomer units in Pluronic F68 (182), we obtain approximately 14.8 Å^2^/monomer. This value is in excellent agreement with those reported in the literature [40].

Further compression of the monolayer is expected to bend the hydrophilic EO chains toward the adjacent liquid phase. This results in a heterogeneous LC1-LC phase, as illustrated in Figure 4. This transition is supported by the surface potential trend, which decreases, likely due to a reduction in the total vertical dipole moment of the polymer units caused by such bending.

When the monolayer is compressed to an area of about 500 Å^2^/molecule at a surface pressure of 23 mNm^−1^, all Pluronic F68 molecules are tightly packed at the air/water interface, with their hydrophilic parts oriented vertically toward the liquid phase. This corresponds to an LC phase. Further compression beyond 500 Å^2^/molecule (23 mNm^−1^) likely leads to monolayer collapse, disrupting its two-dimensional organization and forming three-dimensional polylayer structures. Assuming that the surface area is now occupied only by the 29 PO monomer units of Pluronic F68, the average area per PO monomer unit at the air/water interface can be estimated as 500 Å^2^/molecule divided by 29, yielding approximately 17.2 Å^2^/monomer, which closely matches the value obtained for Pluronic F68 molecules in the LC1 phase.

Observing the progression of the surface pressure/area isotherm during the compression of the Pluronic F68 monolayer, it appears that for areas between 8000 Å^2^/molecule and 6000 Å^2^/molecule, the monolayer remains in the gas phase. Within this range, both the surface pressure remains close to zero, and the surface potential remains steady at around 250 mV. As the monolayer is further compressed past 6000 Å^2^/molecule, a slight increase in surface pressure is observed, becoming more pronounced near 2700 Å^2^/molecule, where the slope of the pressure curve steepens significantly.

Comparing this trend with the surface potential, it is evident that, between the areas of 6000 Å^2^/molecule and 2700 Å^2^/molecule, the surface potential increases due to the freedom of the polymer’s monomer units at the air/water interface, which affects the vertical component of the total dipole moment. Beyond areas of 2000 Å^2^/molecule, the slope of the surface pressure curve increases sharply, reaching up to 35 mNm^−1^. Further compression from an area of 2700 Å^2^/molecule indicates a condensed state of the monolayer, with an inflection point in the isotherm marking the close packing of monomer units. Past this, surface pressure drops sharply, corresponding to the monolayer’s collapse, resulting in the formation of 3D multilayer structures.

It can be concluded that the monolayer remains organized up to a surface pressure of around 10.5 mNm^−1^ and an area of 2700 Å^2^/molecule. Assuming equal surface areas for the EO and PO monomer units of Pluronic F68, the average area per monomer unit at the water/air interface can be estimated by dividing the closely packed area of 2700 Å^2^/molecule by the total 182 monomer units in F68, yielding approximately 14.8 Å^2^/monomer. This result aligns well with previous findings [40].

Figure 5 illustrates the changes in surface pressure (blue curve) and surface potential (red curve) over time for monolayers formed by spreading a suspension of PBCA nanoparticles stabilized with Pluronic F68 at the air/water interface. From the surface pressure vs. time graph, two distinct stages are observed: an initial rapid increase from 0.5 ± 0.2 mNm^−1^ to 1.3 ± 0.2 mNm^−1^ within the first 10 min, followed by a slower increase from 1.3 ± 0.2 mNm^−1^ to 1.8 ± 0.2 mNm^−1^ over the next 110 min. Similarly, in the surface potential vs. time graph, the potential initially rises from 200 ± 10 mV to 250 ± 10 mV within 10 min, remaining constant thereafter.

This kinetic behavior suggests two simultaneous processes following the spreading of the PBCA suspension: an irreversible diffusion of PBCA nanoparticles into the bulk liquid phase and a transformation at the air/water interface. In this second step, the nanoparticles adsorb, and as their Pluronic F68 shells disintegrate, they form a mixed monolayer of bare PBCA nanoparticles and Pluronic F68 molecules at the interface (Figure 5, the inset).

To further investigate the proposed mechanisms of molecular reorganization at the interface, we compared the behavior of a spread suspension of PBCA nanoparticles stabilized with Pluronic F68 to that of Pluronic F68 monolayers. These monolayers were compressed to specific surface pressures, held constant, and monitored for relative changes in area over time.

For instance, Figure 6 shows the relative surface area change over time for both spread PBCA nanoparticles (red curve) and the Pluronic F68 monolayer (blue curve) under barostatic conditions at π = 10 mNm^−1^. The Pluronic F68 monolayer, kept at this constant pressure, showed minimal solubility, with a slight decrease in the relative area to $\frac{∆A(t)}{A_{0}}=-0.05$ after 60 min. In contrast, the spread PBCA nanoparticle suspension, also compressed to π = 10 mNm^−1^, underwent interfacial reorganization, leading to a significant increase in the relative monolayer area to $\frac{∆A(t)}{A_{0}}=0.5$ under the same conditions.

These observations support the suggested molecular mechanism: upon spreading the PBCA dispersion, the interfacial layer likely comprises both the Pluronic molecules that originally formed the nanoparticle shells and the PBCA nanoparticles themselves.

The goal of the present study was to investigate the kinetics and mechanism of interactions of PBCA nanoparticles with a model membrane system, such as the Langmuir monolayer of DPPC, compressed to different states corresponding to specific surface pressures. Experimentally, the monolayer was compressed to a designated pressure, after which the barostat mode was activated. Once the target surface pressure was reached and held constant, a suspension of PBCA nanoparticles stabilized with Pluronic 68 was injected into the reaction compartment of the modified Langmuir trough.

Figure 7a presents the kinetic curves showing changes in relative monolayer area versus time for initial values of π = 5 mNm^−1^, π = 10 mNm^−1^, π = 15 mNm^−1^, and π = 20 mNm^−1^. From the kinetic curves for π = 5 mNm^−1^ and π = 10 mNm^−1^, the observed change in relative area can be formally divided into two phases. The first, characterized by a steeper curve slope, aligns with the accepted molecular model describing PBCA nanoparticle behavior at the air/water interface; it suggests that during this stage, the nanoparticles are adsorbed onto the monomolecular layer of DPPC. The second, slower stage reflects the gradual release and reorganization of Pluronic F68 molecules from the PBCA nanoparticle shells, followed by their incorporation into the monolayer.

Increasing the initial surface pressure leads to smaller relative area changes, indicating less Pluronic 68 penetration. At higher pressures (π = 15 mNm^−1^ and π = 20 mNm^−1^), almost no change in monolayer surface area is observed. This result is consistent with the behavior observed during enzymatic hydrolysis of phospholipid monolayers by phospholipase A_2_ (PLA_2_), where enzyme activity ceases above a certain pressure due to the enzyme’s inability to penetrate a densely packed phospholipid monolayer [26].

In addition to tracking the monolayer area change over time, the ∆V potential was also measured as a function of time. The resulting curves for the respective initial surface pressures of the monolayer are shown in Figure 7b.

At all initial surface pressures, a sharp, jump-like change in potential was observed immediately after nanoparticle injection. This abrupt shift may result from monolayer perturbation during the injection, as surface potential is more sensitive to such disturbances compared to surface pressure. Additionally, it could be attributable to the rapid diffusion of PBCA nanoparticles toward the DPPC monolayer. The largest potential changes were observed at higher initial pressures (π = 15 mNm^−1^ and π = 20 mNm^−1^), with ∆V = 450 mV in both cases. At lower pressures (π = 5 mNm^−1^ and π = 10 mNm^−1^), ∆V values were 315 mV and 420 mV, respectively. Following these initial jumps, only a slight increase in surface potential was noted, likely due to monolayer reorganization. Notably, at the lowest initial surface pressure (π = 5 mNm^−1^), the largest change in surface potential is observed, with its value increasing from ∆V = 315 mV to approximately ∆V = 370 mV, corresponding to ∆(∆V) = 55 mV. In contrast, at higher initial pressures, this change is nearly constant at around ∆(∆V) = 30 mV.

At the end of each experiment involving a DPPC monolayer, in which a suspension of PBCA nanoparticles stabilized with Pluronic 68 was injected into the aqueous support, the films from the water/air interface were vertically transferred to solid mica supports using the Langmuir–Blodgett (LB) method. These films were subsequently topographically scanned with atomic force microscopy (AFM) in semi-contact (tapping) mode.

Today, AFM is firmly established as a reliable method for studying the molecular organization of films obtained via the LB method [41,42]. It provides valuable insights into the lateral organization of various monolayers and their interactions with biologically active molecules at the interface. The AFM visualization of surfaces transferred after the injection of the PBCA suspension corroborated data obtained regarding changes in surface pressure and potential over time (Figure 7).

The AFM images presented in Figure 8 show the topography of the LB films, demonstrating different modes of reorganization and embedding of PBCA nanoparticles into the DPPC monolayers at four different pressures: π = 5 mNm^−1^, π = 10 mNm^−1^, π = 15 mNm^−1^, and π = 20 mNm^−1^. For example, at pressures of π = 5 mNm^−1^ and π = 10 mNm^−1^ (Figure 8a,b), some PBCA nanoparticles are observed integrated into the DPPC monolayer, while areas of the film exhibit distinguishable lateral organization.

At these surface pressures, the DPPC monolayer is either in the homogeneous liquid (L) state or at the boundary between the L and liquid-expanded (LE) states, where regions of heterogeneous separation are not expected; however, the penetration of PBCA nanoparticles and the subsequent decomposition of their shells, releasing Pluronic F68 molecules, could potentially induce phase separation similar to that observed with cholesterol [43]. Furthermore, section analysis of the domain height reveals measurements ranging from 5 to 8 Å. These domains suggest the occurrence of a phase transition in the mixed DPPC-Pluronic 68-PBCA monolayer, likely corresponding to the LE-liquid condensed (LC) transition [44,45]. At surface pressures of π = 15 mNm^−1^ and π = 20 mNm^−1^ (Figure 8c,d), where the monolayer area remains stable over time after the injection of the PBCA nanoparticle suspension, only single nanoparticles are observed on the imaged LB films, and this occurs predominantly at the lower pressure. At the same time, section analysis of specific regions of the film indicates changes in the lateral organization, revealing small-scale structures associated with density and compositional fluctuations (Figure 8d). These fluctuations may be attributable to the partial penetration of Pluronic F68 molecules released from the PBCA nanoparticle shells.

Because AFM phase imaging is conducted in tapping mode, where the phase signal shift is recorded simultaneously with the height signal, we compared the obtained height and phase images to investigate phase separation in the DPPC monolayer further. The AFM images presented in Figure 9 show height and phase contrasts for the lowest and highest surface pressures: π = 5 mNm^−1^ (Figure 9a) and π = 20 mNm^−1^ (Figure 9b), respectively.

It was first observed that the phase signal is highly sensitive to variations in sample composition, adhesion, friction, viscoelasticity, and other factors [46,47]. Garcia et al. [47] demonstrated that the phase signal reflects the energy dissipation involved in tip-sample interactions [48]. As a result, it is commonly assumed that phase contrast reveals adhesion or viscoelastic properties of the studied surfaces; however, understanding the contributions of individual factors to the phase shift remains complex and nontrivial.

In the DPPC monolayer images at π = 5 mNm^−1^ (Figure 9a), the phase contrast image (lower panel in Figure 9a) reveals domains that match the size and shape of those observed in the height image. More intriguingly, the results for DPPC monolayers at the higher surface pressure of π = 20 mNm^−1^ (Figure 9b) display notable differences. Although the differences in height indicating changes in the lateral organization of the DPPC monolayer are subtle and barely discernible in the topography image, these features are profoundly highlighted in the phase image. The phase image not only reveals the presence of phase domains but also clearly distinguishes stripe-like features.

The AFM analysis reveals that PBCA nanoparticles and their Pluronic 68 shell significantly affect the structural and phase properties of DPPC monolayers. These interactions likely occur through particle incorporation or/and Pluronic shell reorganization at the air/water interface. Furthermore, as observed in prior studies on Pluronic 188, it has been shown to insert into DPPC monolayers at surface pressures of approximately 22 mNm^−1^ or lower [49]. This threshold correlates with the maximum surface pressure that P188 can achieve on a pure water subphase, indicating its integration behavior within lipid layers under specific conditions. Comparable effects have been reported for carbon black and silica particles, which alter DPPC monolayer packing by modifying lipid cohesion [50]. The authors of these studies suggested that particle geometry and aggregation at the interface may determine their impact on lipid layer incorporation. Our findings favor similar conclusions; however, further studies involving experimental methods to analyze the mechanical and rheological properties of the monolayers will be necessary to understand these interactions fully.

## 4. Conclusions

Nanoparticles for drug delivery are almost always stabilized with surfactants or other amphiphilic molecules to ensure their colloidal stability in biological fluids; however, it is well known that such surfactants can affect the integrity and permeability of cell membranes, potentially influencing drug penetration into cells. To understand and elucidate the molecular mechanisms behind the interactions of PBCA nanoparticles stabilized with Pluronic 68, we investigated their penetration and reorganization properties upon injecting their suspension into the subphase of a preformed DPPC monolayer. This evaluation was conducted through measurements in a barostatic regime at different surface pressures corresponding to various states of the DPPC monolayer. Our in vitro model, which uses artificial phospholipid membranes and Pluronic-coated PBCA nanoparticles, represents a preliminary step toward understanding the complex interactions within the nanoparticle-surfactant-membrane system.

First, we examined the state of the Pluronic F68 monolayer, in which molecules form a shell around the PBCA nanoparticles and play a crucial role in these molecular processes. We measured surface pressure and surface potential versus area isotherms, estimating the area per molecule of Pluronic F68 and per monomer unit.

From the barostatic regime measurements of the relative surface area and surface potential changes over time following the spreading of the PBCA nanoparticle suspension, we proposed a molecular mechanism for their reorganization. The spreading of the nanoparticles exhibits two-stage kinetics: in the first stage, the nanoparticles undergo rapid adsorption at the interface, followed by a slower molecular reorganization that involves the disruption and release of the stabilizing Pluronic F68 shell.

We also utilized AFM to visualize the state of the DPPC at the completion of the kinetic process involving the penetration and reorganization of the PBCA nanoparticles. Future perspectives for elucidating the proposed mechanism and developing a quantitative model include conducting experiments with different suspension concentrations of PBCA nanoparticles and incorporating other experimental techniques, such as Brewster angle microscopy or epi-fluorescent microscopy [40,51].

## Figures and Tables

**Figure 1 membranes-14-00254-f001:**
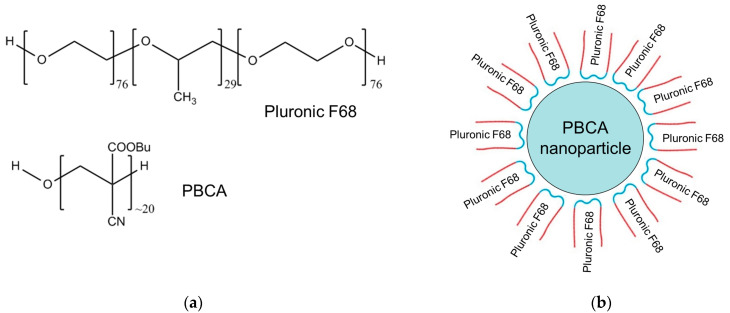
(**a**) Structural formulas of Pluronic F68 (poloxamer 188) and PBCA, along with (**b**) a schematic illustration of the possible organization of poloxamer molecules forming a shell around a PBCA nanoparticle.

**Figure 2 membranes-14-00254-f002:**
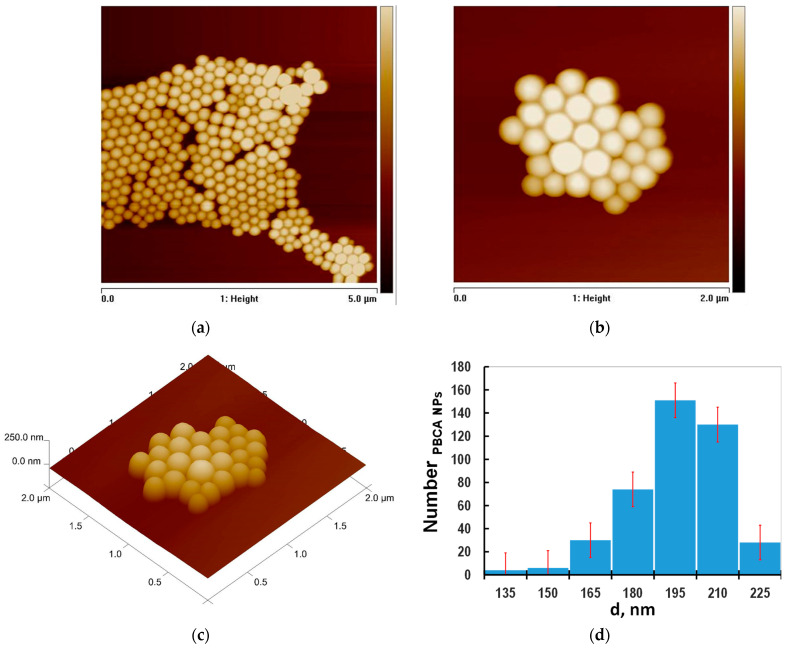
AFM images of PBCA NPs deposited on freshly cleaved mica. (**a**) Scanned surface area- $5\times 5\;{\mu m}^{2}$ and scale at z = 300 nm (**b**) $2\times 2\;{\mu m}^{2}$, z-scale is z = 250 nm and corresponding (**c**) 3d image of the surface (**d**) Histogram of the PBCA nanoparticles size distribution.

**Figure 3 membranes-14-00254-f003:**
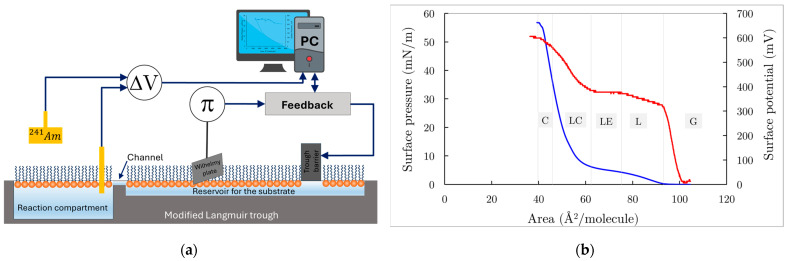
(**a**) A modified Langmuir bath has a reaction compartment with area $A_{0}=50\;{\mathrm{cm}}^{2}$, which is interconnected with the reservoir compartment by two narrow channels. The surface pressure is measured by the Wilhelmy method and is kept constant by the feedback controller unit, which regulates the trough’s barrier movement so that π = const. (**b**) Surface pressure vs. Area (blue curve) and Surface potential vs. Area (red curve) isotherms. On both isotherms clearly are distinguished gaseous (G), liquid expanded (LE), liquid condensed (LC) and condensed states of DPPC monolayer as described elsewhere [34].

**Figure 4 membranes-14-00254-f004:**
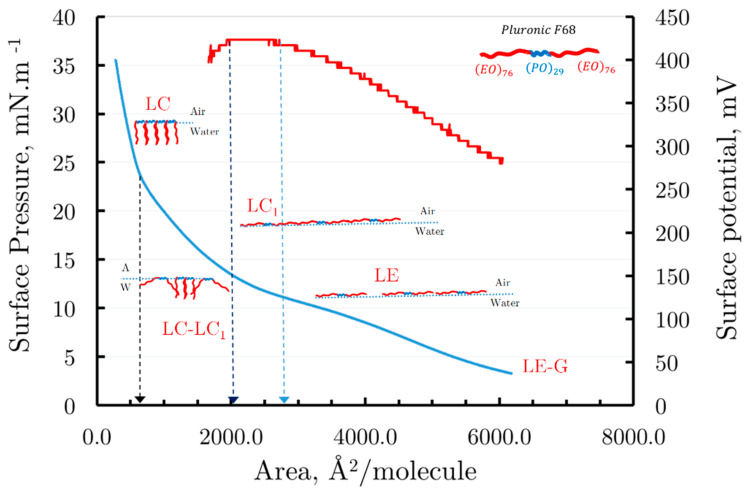
The surface pressure–area (blue curve) and surface potential–area isotherms (red curve, the secondary ordinate) isotherms of Pluronic F68 spread at the air/water interface. The arrows indicate the areas corresponding to the thermodynamic states that the Pluronic 68 monolayer transitions through during compression.

**Figure 5 membranes-14-00254-f005:**
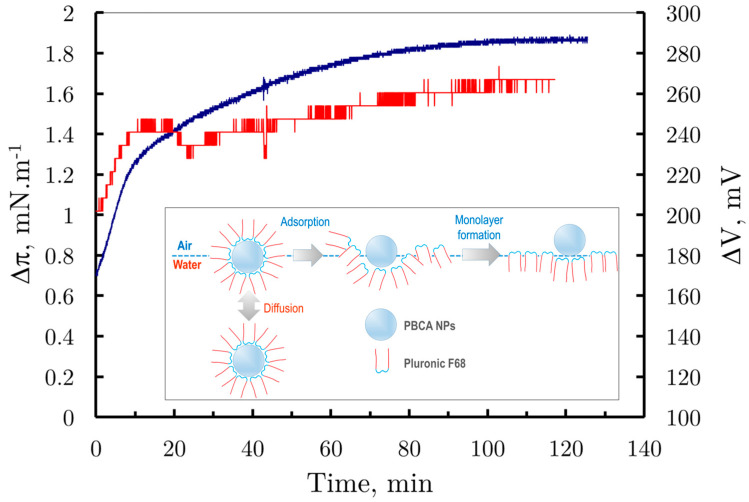
Comparison of surface pressure changes vs. time (blue curve) and surface potential vs. time (red curve, secondary ordinate) of PBCA nanoparticles spread at the air/water interface from water solution. *The inset*: Suggested molecular mechanism of the nanoparticles’ reorganization at the interface.

**Figure 6 membranes-14-00254-f006:**
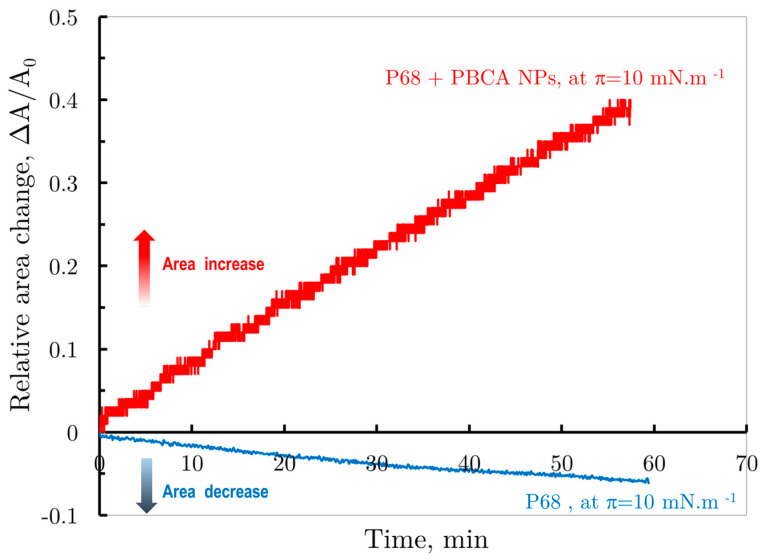
Comparison of the change in relative monolayer of suspension of Pluronic 68 stabilized PBCA nanoparticles (red curve) and monolayer of Pluronic F68 (blue curve) versus in barostatic regime at constant surface pressure π = 10 mNm ^−1^.

**Figure 7 membranes-14-00254-f007:**
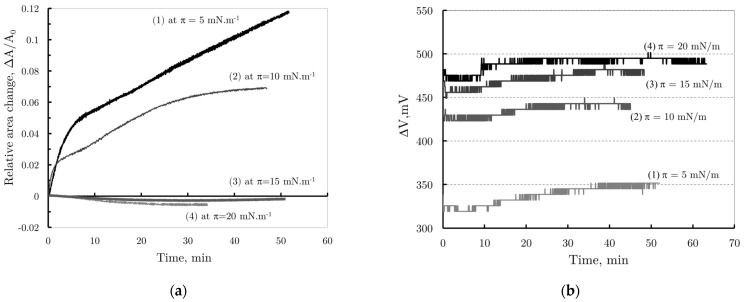
(**a**) The relative area changes and (**b**) the change in Surface potential vs. tame of DPPC monolayers after injection of a PBCA nanoparticles suspension stabilized with Pluronic 68, in barostatic mode, at pressures π = 5 mNm ^−1^, π = 10 mNm ^−1^, π = 15 mNm ^−1^, and π = 20 mNm ^−1^, respectively.

**Figure 8 membranes-14-00254-f008:**
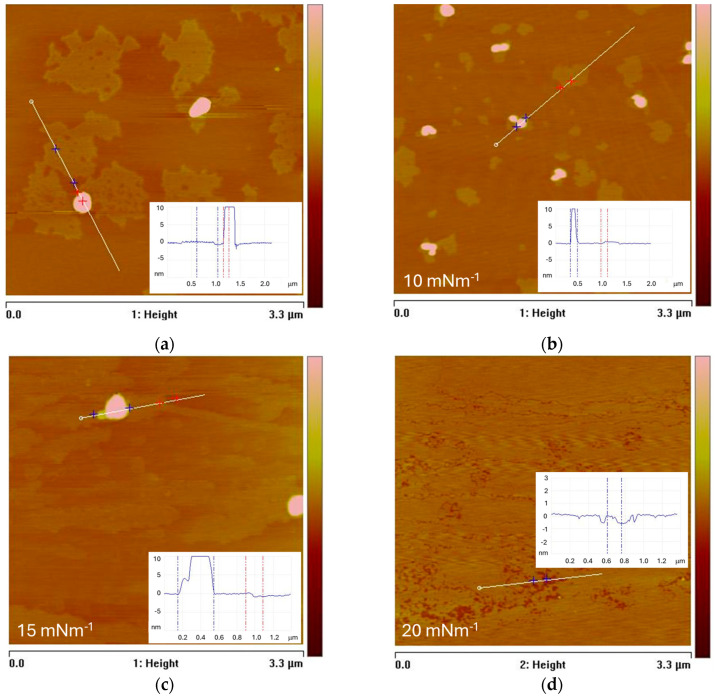
AFM images in 2D format of the topography of LB films transferred at the end of the kinetic and processes of molecular reorganization that occurred after injection beneath the DPPC monolayer of suspension of PBCA nanoparticle at the initial surface pressures: (**a**) 5 mNm^−1^; (**b**) 10 mNm^−1^; (**c**) 15 mNm^−1^; (**d**) 20 mNm^−1^. All the scanned areas of the images are 3.3 × 3.3 μm^2^ with a z-range of 15 nm for (A–C) and 5 nm for (D). Each image is accompanied by cross-section curves corresponding to the lines marked on the images.

**Figure 9 membranes-14-00254-f009:**
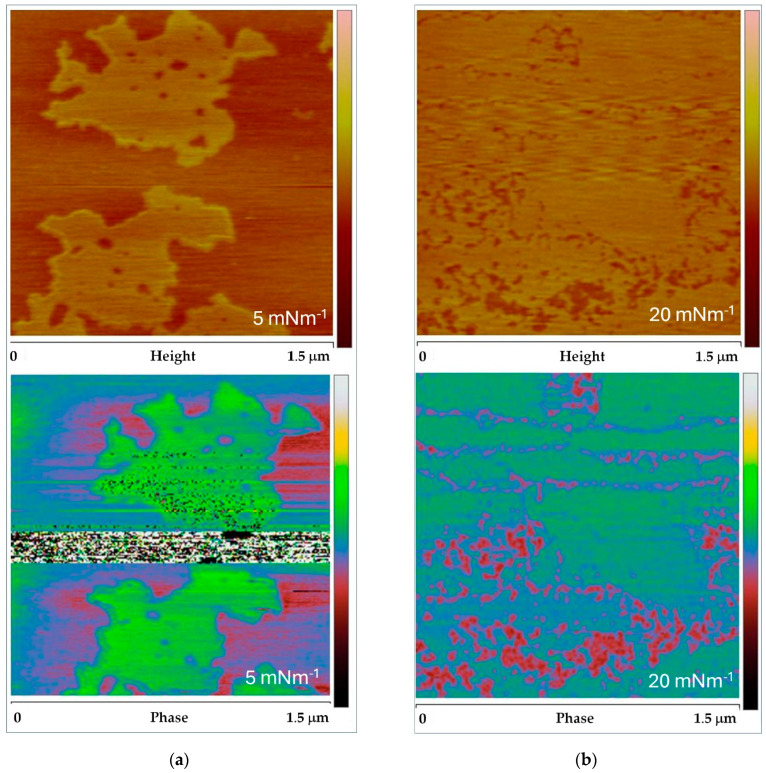
Height (upper) and phase (lower) AFM images of a DPPC monolayer illustrate heterogeneity and lateral phase segregation at surface pressures of (**a**) 5 mNm ^−1^ and (**b**) 20 mNm ^−1^. All images are 1.5 × 1.5 μm^2^, with a z-range of 5 nm for the height images and 45° for the phase images. *Note*: The phase images are presented as raw data without any filtering, which results in some noise signals, particularly the stripe visible in the middle of the phase image in (**a**).

## Data Availability

Data are available upon request from the authors.

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
