# Peer review of "The Langmuir Monolayer as a Model Membrane System for Studying the Interactions of Poly(Butyl Cyanoacrylate) Nanoparticles with Phospholipids at the Air/Water Interface"

_membranes, 2024, doi:10.3390/membranes14120254_

Round 1
Reviewer 1 Report
Comments and Suggestions for Authors
The paper presents a multi-èerspective investigation aimed at unveiling the mechanisms of interaction of Poly(butyl cyanoacrylate) nanoparticles with membranes.
The work nicely connects morphology information obatined by AFM to observation obtained by Langmuir films on lateral pressure and surface potentials.
In general I find the work is well conducted and conclusions are sound.
Still, I have a few comments:
1) Can authors quantify the number of repetitions performed for each presented experiment? Saying many times is not enough.
2) In the introduction it is written that DPPC is the most common phospholipid of membranes. This is not true if not for lungs. Citations, indeed, for such statement, are not appropriate and do not refer to biological studies. Authors should deepen this aspect: for themselves and to set up a proper discussion in the introduction clearly explaining why their model is appropriate and for what.
3) Fig 1.b: it is not clear if the NP components organization is guessed from the chemical properties of components or if it is assessed by some experiments.
4) Figure 4: data collection starts from an area of 6000 A/mol, while authors discuss results form 800 A/mol. Even if I also espect the system to be in G phase at bigger areas and the surface potential to be not higher than 250 mV, it is not possible to discuss not existent data. The discussion must refer to data presented.
5) It is not clear along the text and in the conclusion what is the interest of authors findings, in connection to drug/gene/... delivery needs. Also seen that the investigated NPs are big and really at the limit of a nanodelivery system. Therefore I think it is important to dentify what they can be exploited for, once (even partially) understood the membrane interaction mechanism.
Author Response
Response to Review 1
First, we would like to thank Reviewer 1 for the valuable comments, remarks, and suggestions. We have carefully considered all of them point by point and revised the text accordingly. We now hope that this version of the manuscript is ready for approval and can be published.
Reviewer 1: The paper presents a multi-èerspective investigation aimed at unveiling the mechanisms of interaction of Poly(butyl cyanoacrylate) nanoparticles with membranes.The work nicely connects morphology information obtained by AFM to observation obtained by Langmuir films on lateral pressure and surface potentials. In general I find the work is well conducted and conclusions are sound.
Still, I have a few comments:
- Can authors quantify the number of repetitions performed for each presented experiment? Saying many times is not enough.
Answer: We appreciate the referee’s remark and have incorporated the required information in the following line: 'Under these conditions, the isotherms were highly reproducible. At least three measurements were performed for this and all other experiments in the study to confirm reproducibility.
- Reviewer 1: In the introduction it is written that DPPC is the most common phospholipid of membranes. This is not true if not for lungs. Citations, indeed, for such statement, are not appropriate and do not refer to biological studies. Authors should deepen this aspect: for themselves and to set up a proper discussion in the introduction clearly explaining why their model is appropriate and for what.
Answer: As the referee suggested we added a proper explanation in the introduction and references that support it.
- Reviewer 1: Fig 1.b: it is not clear if the NP components organization is guessed from the chemical properties of components or if it is assessed by some experiments.
Answer: This is an important remark, and as the referee suggests, it requires clarification. Indeed, this molecular configuration is speculative and primarily based on the chemical properties of the components rather than direct experimental evidence. We propose such a structure because PBCA nanoparticles are inherently hydrophobic. As a surfactant, Pluronic F68 acts as a stabilizing agent and is expected to form a shell around individual nanoparticles. In this configuration, the hydrophobic regions of Pluronic F68 interact with the nanoparticle surface, while the hydrophilic regions face the aqueous phase.
We believe this model is plausible. Assuming that Pluronic F68 adsorption on the nanoparticle surface is similar to its behavior at the air-water interface (as discussed further), we can estimate the number of molecules involved. Given a mean nanoparticle radius of approximately 100 nm (1000 â„«), the surface area of a single PBCA nanoparticle can be calculated as SNP=4πr2=1.25*107 â„«2. For a closely packed monolayer of Pluronic F68 molecules at the air-water interface, each molecule occupies an area of S Pluronic F68=500 â„«2 per molecule per molecule, Thus, the number of Pluronic F68 molecules that a single PBCA nanoparticle can accommodate is estimated as: n= SNP / S Pluronic F68=25000 Pluronic F68 molecules.
- Reviewer 1: Figure 4: data collection starts from an area of 6000 A/mol, while authors discuss results form 800 A/mol. Even if I also expect the system to be in G phase at bigger areas and the surface potential to be not higher than 250 mV, it is not possible to discuss not existent data. The discussion must refer to data presented.
Answer: We are grateful for the referee’s remark and agree that discussions should be based solely on the presented data. Furthermore, recognizing the need for a more in-depth analysis of the Pluronic F68 isotherms, we have provided a revised description of the events occurring at the air/water interface during the compression of the Pluronic F68 spread monolayer.
Our experimental data start from areas of 6500 Ų/molecule, tracking the progression of the surface pressure/area isotherm curve during the compression of the Pluronic 68 monolayer to areas as small as 500 Ų/molecule. Technically, we were able to monitor the surface potential only down to areas of about 1800 Ų/molecule. However, using previously published data, it can be inferred that for areas between 8000 Ų/molecule and 6500 Ų/molecule, the monolayer is most likely in a gaseous-liquid expanded (G-LE) state, as schematically illustrated in Fig. 4. During this range, the surface pressure gradually and slowly increases, while the surface potential also rises due to the freedom of the polymer's monomer units at the air/water interface, which contribute to changes in the vertical component of the total dipole moment.
Further compression of the monolayer from 6500 Ų/molecule to approximately 3000 Ų/molecule results in the appearance of a liquid-expanded (LE) phase, followed by a sloping transition plateau that corresponds to a liquid-condensed 1 (LC1) phase. Strong evidence for this monolayer state is provided by the surface potential isotherm, which exhibits a plateau in this region. In this state, as shown in Fig. 4, the polymer molecules of Pluronic F68 are closely packed at the air/water interface.
At a surface pressure of approximately 10.5 mNm-1 and a corresponding area of 2700 Ų/molecule, assuming that the surface areas occupied by the ethylene oxide (EO) and propylene oxide (PO) monomer units of Pluronic F68 are approximately equal, the average area per monomer unit can be estimated. Dividing the area of the closely packed monolayer (2700 Ų/molecule) by the total number of monomer units in F68 (182), we obtain approximately 14.8 Ų/monomer. This value is in excellent agreement with those reported in the literature.
Further compression of the monolayer is expected to bend the hydrophilic EO chains toward the adjacent liquid phase. This results in a heterogeneous LC1-LC phase, as illustrated in Fig. 4. This transition is supported by the surface potential trend, which decreases, likely due to a reduction in the total vertical dipole moment of the polymer units caused by such bending.
When the monolayer is compressed to an area of ~ 500 Ų/molecule at a surface pressure of 23 mNm-1, all Pluronic F68 molecules are tightly packed at the air/water interface, with their hydrophilic parts oriented vertically toward the liquid phase. This corresponds to an LC phase. Further compression beyond 500 Ų/molecule (23 mNm-1) likely leads to monolayer collapse, disrupting its two-dimensional organization and forming three-dimensional polylayer structures. Assuming that the surface area is now occupied only by the 29 PO monomer units of Pluronic F68, the average area per PO monomer unit at the air/water interface can be estimated as 500 Ų/molecule divided by 29, yielding approximately 17.2 Ų/monomer. This result closely matches the value obtained for Pluronic F68 molecules in the LC1 phase.
- Reviewer 1: It is not clear along the text and in the conclusion what is the interest of authors findings, in connection to drug/gene/... delivery needs. Also seen that the investigated NPs are big and really at the limit of a nanodelivery system. Therefore I think it is important to identify what they can be exploited for, once (even partially) understood the membrane interaction mechanism.
Answer: This is an important remark made by the referee and must be carefully considered. Nanoparticles for drug delivery are almost always stabilized with surfactants or other amphiphilic molecules to ensure their colloidal stability in biological fluids. However, it is well known that such surfactants can affect the integrity and permeability of cell membranes, potentially influencing drug penetration into cells. Our in vitro model, which uses artificial phospholipid membranes and Pluronic-coated PBCA nanoparticles, represents a preliminary step toward understanding the complex interactions within the nanoparticle-surfactant-membrane system.
Reviewer 2 Report
Comments and Suggestions for Authors
Report on the paper « The Langmuir monolayer as a model membrane system for studying the interactions of poly(butyl cyanoacrylate) nanoparticles with phospholipids at the air/water interface » by Georgi Yordanov et al
The paper presents a study on DPPC Langmuir film deposited on a subphase containing PBCA nanoparticles (NPs) stabilized by Pluronic F68 triblock polymer. The system, used to mimic the interactions between PBCA NPs and the cell membrane, has been studied on liquid subphase by Surface pressure versus Surface density isotherms and surface potential and by AFM microscopy after transfer on solid substrate. The study is interesting and deserves to presented. However, although the experiments appear to be correctly proceeded, I disagree with the analysis of the results and consequently on the conclusion.
My first concern is about the description/interpretation of the isotherms:
Figure 4 presents the isotherm of Pluronic F68. I would interpret the curve as follow. A G-LE transition plateau for area larger than 6500 A2/mol (not presented), a LE phase from 6500 A2/mol up to about 3000 A2/mol, then, a sloping transition plateau to a LC1 phase from this value up to about 2000 A2/mol and at least a LC1-LC transition at about 500 A2/mol (23 mN/m). The authors propose a collapse at 3500 A2/mol which is really not obvious considering the figure.
Furthermore, since the surface pressure is the difference between the surface tension of the uncovered subphase (gamma0) and that of the subphase covered by the film, did the authors apply the correction resulting from the variation of gamma0 when the NPs are introduced into the subphase when they follow the insertion of the NPs into the monolayer (figure 5 &7)?
Incidentally, about the surface potential measurements, it is mentioned (line 348) “At lower pressures (π = 5 mNm-1 and π = 10 mNm-1), ΔV values were 315 mV and 420 mV, respectively”. Considering the values, one should gather10 mN.m-1 to 15 mN.m-1 and 20 mN.m-1 rather than 5 mN.m-1.
My second concern is about the AFM images.
At 5 and 10 mN/m, it is observed that the NPs modify the thickness of the film surrounding them. At higher pressures (15 and 20 mN/m), very few NPs are observed but the background, representing the organic film, presents inhomogeneities. There is clearly a lack of height profiles in different areas to be able to compare the situations.
As a conclusion, the analysis of the results is over-interpreted. As a consequence, the manuscript cannot be published in its present form.

Author Response
Response to Review 2
First, we would like to thank Reviewer 2 for the valuable comments, remarks, and suggestions. We have carefully considered all of them point by point and revised the text accordingly. We now hope that this version of the manuscript is ready for approval and can be published.
Reviewer 2: The paper presents a study on DPPC Langmuir film deposited on a subphase containing PBCA nanoparticles (NPs) stabilized by Pluronic F68 triblock polymer. The system, used to mimic the interactions between PBCA NPs and the cell membrane, has been studied on liquid subphase by Surface pressure versus Surface density isotherms and surface potential and by AFM microscopy after transfer on solid substrate. The study is interesting and deserves to presented. However, although the experiments appear to be correctly proceeded, I disagree with the analysis of the results and consequently on the conclusion.
- My first concern is about the description/interpretation of the isotherms: Figure 4 presents the isotherm of Pluronic F68. I would interpret the curve as follow. A G-LE transition plateau for area larger than 6500 A2/mol (not presented), a LE phase from 6500 A2/mol up to about 3000 A2/mol, then, a sloping transition plateau to a LC1 phase from this value up to about 2000 A2/mol and at least a LC1-LC transition at about 500 A2/mol (23 mN/m). The authors propose a collapse at 3500 A2/mol which is really not obvious considering the figure.
Answer: We thank the referee for this valuable comment, and, after careful consideration of all available experimental data, we fully agree with the conclusions drawn. As suggested, we have provided a revised description of the events occurring at the air/water interface during the compression of the Pluronic F68 spread monolayer.
Our experimental data start from areas of 6500 Ų/molecule, tracking the progression of the surface pressure/area isotherm curve during the compression of the Pluronic 68 monolayer to areas as small as 500 Ų/molecule. Technically, we were able to monitor the surface potential only down to areas of about 1800 Ų/molecule. However, using previously published data, it can be inferred that for areas between 8000 Ų/molecule and 6500 Ų/molecule, the monolayer is most likely in a gaseous-liquid expanded (G-LE) state, as schematically illustrated in Fig. 4. During this range, the surface pressure gradually and slowly increases, while the surface potential also rises due to the freedom of the polymer's monomer units at the air/water interface, which contribute to changes in the vertical component of the total dipole moment.
Further compression of the monolayer from 6500 Ų/molecule to approximately 3000 Ų/molecule results in the appearance of a liquid-expanded (LE) phase, followed by a sloping transition plateau that corresponds to a liquid-condensed 1 (LC1) phase. Strong evidence for this monolayer state is provided by the surface potential isotherm, which exhibits a plateau in this region. In this state, as shown in Fig. 4, the polymer molecules of Pluronic F68 are closely packed at the air/water interface.
At a surface pressure of approximately 10.5 mN.m-1 and a corresponding area of 2700 Ų/molecule, assuming that the surface areas occupied by the ethylene oxide (EO) and propylene oxide (PO) monomer units of Pluronic F68 are approximately equal, the average area per monomer unit can be estimated. Dividing the area of the closely packed monolayer (2700 Ų/molecule) by the total number of monomer units in F68 (182), we obtain approximately 14.8 Ų/monomer. This value is in excellent agreement with those reported in the literature.
Further compression of the monolayer is expected to bend the hydrophilic EO chains toward the adjacent liquid phase. This results in a heterogeneous LC1-LC phase, as illustrated in Fig. 4. This transition is supported by the surface potential trend, which decreases, likely due to a reduction in the total vertical dipole moment of the polymer units caused by such bending.
When the monolayer is compressed to an area of about 500 Ų/molecule at a surface pressure of 23 mN.m-1, all Pluronic F68 molecules are tightly packed at the air/water interface, with their hydrophilic parts oriented vertically toward the liquid phase. This corresponds to an LC phase. Further compression beyond 500 Ų/molecule (23 mN.m-1) likely leads to monolayer collapse, disrupting its two-dimensional organization and forming three-dimensional polylayer structures. Assuming that the surface area is now occupied only by the 29 PO monomer units of Pluronic F68, the average area per PO monomer unit at the air/water interface can be estimated as 500 Ų/molecule divided by 29, yielding approximately 17.2 Ų/monomer. This result closely matches the value obtained for Pluronic F68 molecules in the LC1 phase.
- Reviewer 2: Furthermore, since the surface pressure is the difference between the surface tension of the uncovered subphase (gamma0) and that of the subphase covered by the film, did the authors apply the correction resulting from the variation of gamma0 when the NPs are introduced into the subphase when they follow the insertion of the NPs into the monolayer (figure 5 &7)?
Answer: We thank the referee for this comment. Indeed, surface pressure is defined as the difference between the surface tension of the pure subphase (γ0) and that of the monolayer-covered surface. In the experiments corresponding to Figures 5 and 7, we injected the nanoparticle solution beneath the preformed DPPC monolayer at the specified surface pressure. Subsequently, we activated the barostat to maintain constant surface pressure and monitored the changes in surface area over time. This approach ensured that any observed changes could be attributed to molecular events occurring within the phospholipid monolayer and its reorganization at the air/water interface.
- Reviewer 2: Incidentally, about the surface potential measurements, it is mentioned (line 348) “At lower pressures (π = 5 mNm-1 and π = 10 mNm-1), ΔV values were 315 mV and 420 mV, respectively”. Considering the values, one should gather10 mN.m-1to 15 mN.m-1 and 20 mN.m-1 rather than 5 mN.m-1.
Answer: We acknowledge the referee's observation of the surface potential measurements on line 348. Upon reviewing the data and the ΔV values, it is indeed appropriate to reference the surface pressures of 10 mN.m-1, 15 mN.m-1, and 20 mN.m-1, in addition to the data for the surface pressure of 5 mN.m-1. We have corrected this in the revised manuscript.
- Reviewer 2: My second concern is about the AFM images. At 5 and 10 mN/m, it is observed that the NPs modify the thickness of the film surrounding them. At higher pressures (15 and 20 mN/m), very few NPs are observed but the background, representing the organic film, presents inhomogeneities. There is clearly a lack of height profiles in different areas to be able to compare the situations.
Answer: We thank the referee for this valuable remark. After carefully reviewing the available images and presenting them alongside the section analysis, we have addressed the referee’s suggestion accordingly.
Reviewer 3 Report
Comments and Suggestions for Authors
1. More details regarding the synthesis of butyl cyanoacrylate (BCA) monomer should be provided in the Materials section.
2. References for preparing colloidal PBCA nanoparticles can cite recent literature.
3. Figure 1. (b) is very poor, replace it with legible structures.
4. Please adjust the font size of Formula 1 to the same size.
Author Response
Response to Review 3
First, we would like to thank Reviewer 3 for the valuable comments, remarks, and suggestions. We have carefully considered all of them point by point and revised the text accordingly. We now hope that this version of the manuscript is ready for approval and can be published.
- Reviewer 3: More details regarding the synthesis of butyl cyanoacrylate (BCA) monomer should be provided in the Materials section.
Answer: The BCA monomer used in our synthesis was a commercially available product obtained from a local production company, Special Polymers Ltd. (Bulgaria). The detailed production technology of the monomer is proprietary and not disclosed to customers."
- Reviewer 3: References for preparing colloidal PBCA nanoparticles can cite recent literature.
Answer: The second paragraph of the Introduction has been rewritten, and new references on PBCA nanoparticles have been included in the revised manuscript.
- Reviewer 3: Figure 1. (b) is very poor, replace it with legible structures.
Answer: The old figure has been replaced with an improved version featuring clearer structures of Pluronic F68 and PBCA, along with an enhanced model of the Pluronic F68-coated PBCA nanoparticle
- Reviewer 3: Please adjust the font size of Formula 1 to the same size.
Answer: The font size has been adjusted to ensure consistency in the revised figure (see the response to Q3).
Round 2
Reviewer 1 Report
Comments and Suggestions for Authors
Authors replied to all the doubts raised and I believe that now the manuscript is in an appropriate form for publication.
Author Response
We sincerely thank Reviewer 1 for their valuable efforts and constructive feedback, which have significantly enhanced the quality of our manuscript. We also appreciate their approval and acceptance of the manuscript in its final form.
Reviewer 2 Report
Comments and Suggestions for Authors
Line 318 « in a gaseous-liquid expanded (G-LE) transition » (not “state”)
I still don't really understand the analysis of AFM images, which I find rather summary (no discussion on the phase , sometimes inhomogeneous background which raises questions about the presence of a monolayer over the entire image, etc.).
Author Response
First, we would like to sincerely thank Reviewer 2 for their efforts, insightful remarks, and constructive suggestions, which have significantly contributed to improving the quality of our manuscript.
We have addressed the technical error pointed out by Reviewer 2 regarding Line 318, where we corrected the term "in a gaseous-liquid expanded (G-LE) transition" (not "state").
Additionally, we carefully considered Reviewer 2's comment: “I still don't really understand the analysis of AFM images, which I find rather summary (no discussion on the phase, sometimes inhomogeneous background which raises questions about the presence of a monolayer over the entire image, etc.).”
To further clarify the interpretation of our AFM data, we have included a new Figure 9, which presents images of the phase signal shift recorded simultaneously with the height signal. This addition enables a more comprehensive comparison of the height and phase images, providing further evidence to support our conclusions about phase separation in the DPPC monolayer.
Although the interpretation of phase shifts remains complex and involves contributions from various physical factors, it is well-documented that phase signals are highly sensitive to variations in sample composition, adhesion, friction, viscoelasticity, and other properties. Phase contrast is commonly understood to reflect adhesion or viscoelastic properties of the studied surfaces.
In our experiments, the DPPC monolayer images at π = 5 mN/m (Fig. 9a) show that the phase contrast image (lower panel in Fig. 9a) reveals domains that correspond to the size and shape observed in the height image. More intriguingly, at the higher surface pressure of π = 20 mN/m (Fig. 9b), the phase contrast reveals notable differences. While the differences in height—indicative of changes in the lateral organization of the DPPC monolayer—are subtle and barely discernible in the topography image, they are profoundly highlighted in the phase image. The phase image not only reveals phase domains but also distinctly highlights stripe-like features.
We hope that the revisions and additional data included in this version of the manuscript address the reviewers' concerns and that it will now be considered for approval and publication.
Reviewer 3 Report
Comments and Suggestions for Authors
Accept in present form.
Author Response
We sincerely thank Reviewer 3 for their valuable efforts and constructive feedback, which have significantly enhanced the quality of our manuscript. We also appreciate their approval and acceptance of the manuscript in its final form.